# Direct observation of imploded core heating via fast electrons with super-penetration scheme

T. Gong[1,5], H. Habara[1]*, K. Sumioka[1], M. Yoshimoto[1], Y. Hayashi[1], S. Kawazu[1], T. Otsuki[1], T. Matsumoto[1], T. Minami[1], K. Abe[1], K. Aizawa[1], Y. Enmei[1], Y. Fujita[1], A. Ikegami[1], H. Makiyama[1], K. Okazaki[1], K. Okida[1], T. Tsukamoto[1], Y. Arikawa[2], S. Fujioka [2], Y. Iwasa [2], S. Lee [2], H. Nagatomo[2], H. Shiraga[2], K. Yamanoi[2], M.S. Wei[3] & K.A. Tanaka[1,4]*

Fast ignition (FI) is a promising approach for high-energy-gain inertial confinement fusion in the laboratory. To achieve ignition, the energy of a short-pulse laser is required to be delivered efficiently to the pre-compressed fuel core via a high-energy electron beam. Therefore, understanding the transport and energy deposition of this electron beam inside the pre-compressed core is the key for FI. Here we report on the direct observation of the electron beam transport and deposition in a compressed core through the stimulated Cu K$\alpha$ emission in the super-penetration scheme. Simulations reproducing the experimental measurements indicate that, at the time of peak compression, about 1% of the short-pulse energy is coupled to a relatively low-density core with a radius of 70 μm. Analysis with the support of 2D particle-in-cell simulations uncovers the key factors improving this coupling efficiency. Our findings are of critical importance for optimizing FI experiments in a super-penetration scheme.

[1] Graduate School of Engineering, Osaka University, 2-1 Yamada-oka, Suita, Osaka, 565-0871, Japan. [2] Institute of Laser Engineering, Osaka University, 2-6 Yamada-okaSuitaOsaka 565-0871, Japan. [3] Laboratory for Laser Energetics, University of Rochester, Rochester, NY 14623-1299, USA. [4] Extreme Light Infrastructure: Nuclear Physics, 30 Reatorului, Magurele-Bucharest 077125, Romania. [5] Present address: Laser Fusion Research Center, China Academy of Engineering Physics, Mianyang, Sichuan 621900, People's Republic of China. *email: habara@eei.eng.osaka-u.ac.jp; kazuo.tanaka@eli-np.ro

Energetic electron beams produced by the laser-plasma interaction are useful sources for particle acceleration[1], novel radiation[2], medical therapy[3], the formation of warm dense matter of interest in planetary science[4,5] and astrophysics[6,7], and particularly the development of high-gain laser-driven inertial confinement fusion (ICF)[8,9].

Conventionally, laser-driven ICF is supposed to be achieved via the central hotspot approach. By compressing a deuterium-tritium (D-T) fuel shell either directly with laser beams or indirectly with laser-produced X-rays, a high-temperature hotspot is formed, igniting the nuclear fusion reactions and then burning the surrounding high-density fuel by α-particle heating[10]. This hotspot is formed at the target center by accurately timed shock waves[10], which thereby requires excellent spherical compression symmetry for the ignition, large driver energy and little mixing of surrounding cold fuel into the hotspot. In recent experiments performed on the National Ignition Facility (NIF) at Lawrence Livermore National Laboratory[11], fuel gain exceeding unit has been demonstrated[12], indicating a landmark step towards ignition. However, ignition has not been achieved so far in this central hotspot approach.

As an alternative approach, fast ignition[13] separates the hotspot formation from the compression phase by taking advantage of an additional energetic electron beam, therefore relaxing the requirements of compression symmetry and driver energy. Specifically, the fuel shell is compressed isochorically to a high areal density, and then a fast (relativistic) electron beam produced by an ultra-intense laser (UIL) is injected to rapidly heat the compressed fuel, creating a hotspot[13]. Due to the isochoric compression, this approach allows more fuel to be compressed, thus promising a higher gain than the central hotspot approach.

A key issue in this fast ignition approach is the energy coupling efficiency from the UIL to the compressed core[13]. One solution is to introduce a guide cone into the fuel shell in order to keep the UIL path free from the surrounding plasmas and bring the intrinsically divergent fast electrons close (~50 μm) to the core. The early experiments[8,9] performed in this cone-in-shell scheme at the Institute of Laser Engineering (ILE) in Osaka University reported a coupling efficiency of 15–30%, which was, however, demonstrated to be an optimistic result by the subsequent experiments. Lower coupling efficiencies of ~1.6% and (3.5 ± 1.0)% were, respectively, inferred in the follow-up experiments[14] on the same facility at ILE and in the experiments[15] performed on the OMEGA Laser Facility at the Laboratory for Laser Energetics (LLE). Improvement of the coupling efficiency requires detailed information about the spatial transport of the fast electrons, which motivated the more recent experiments conducted on OMEGA by Jarrott et al.[16,17]. Energy deposition of fast electrons in the compressed cone-in-shell target was spatially identified by exploiting the Kα emission from a Cu tracer doped in the fuel shell. The results suggested that the coupling efficiency was limited by the intrinsic properties of the fast electron source, which not only had a large divergence angle but also was far away from the core due to preplasma filling the cone. In subsequent optimized experiments with a wider cone tip and a high-contrast short pulse, the coupling efficiency was improved to 7% (ref. [16]), making this cone-in-shell scheme a promising way for fast ignition. Further optimization of the experiments has been proposed by creating a collimated electron beam with magnetic fields produced either externally[18,19] or using an engineering target[20,21].

Another unique solution to improve the coupling efficiency is the super-penetration scheme[22–24], proposed based on the original idea of FI, where two successive short pulse UILs are injected into the pre-compressed fuel core: the first UIL producing a low-density plasma channel through the coronal region, while the second, guided by the preformed channel, acting as an ignitor pulse to generate the fast electrons. The idea of the super-penetration scheme is to apply effects such as relativistic self-focusing[25], relativistic induced transparency[26] and hole-boring[27] to the ignitor pulse, so that it can propagate into the overcritical region (where $n_e > n_c$, with $n_e$ and $n_c$, respectively, being the local electron density and the classic critical density), minimizing the transport distance of the fast electrons to the core and hence improving the coupling efficiency. Due to the absence of the cone in the target, the cone-induced issues, such as implosion symmetry breakage, preplasma filling in the cone[16], and mixing of cone material into the fuel core[28] could be avoided in this super-penetration scheme. Because of its simplicity, this scheme appears even advantageous for a future laser fusion reactor. With a planar target, recent experiments on OMEGA have demonstrated the formation of a plasma channel up to overcritical density under fast-ignition-relevant conditions[29] and observed collimated fast electrons on the channel axis[30], rather promising to achieve fast ignition with this super-penetration scheme.

In this article, we report on the integral fast ignition experiments with the super-penetration scheme, which are performed by injecting a short pulse UIL into a spherically compressed target. The transport and energy deposition of fast electrons in the compressed core plasmas are directly observed through the Cu Kα emission induced by fast electrons when interacting with the Cu tracer. An energy coupling efficiency of (0.8 ± 0.3)% from the UIL to the low-density (~0.06 g cm⁻²) core plasma at the time of peak compression is obtained based on the measured Cu Kα photons. Key factors for achieving higher coupling efficiency are identified with the support of 2D particle-in-cell (PIC) simulations for the future ignition experiments. It indicates that the energy coupling efficiency can be improved up to 12% by optimizing the key factors in future ignition-scale plasmas.

## Results

**Experimental setup**. The experiments (see Fig. 1a) were performed on the GEKKO-LFEX laser facility at the Institute of Laser Engineering in Osaka University. The target (see Methods section for more details) used in the experiments consisted of a solid Cu contained CH sphere[31] and a CH (parylene) coating layer, which aimed to prevent the Cu atoms from directly interacting with the lasers. Twelve 526-nm GEKKO-XII (GXII) lasers, with an energy of 200 J per beam and a pulse duration of 1.6 ns in full width at half maximum (FWHM), were used to uniformly compress the target. A typical pulse shape of GXII lasers is shown in Fig. 1b, with $t = 0$ ns defined as 2.0 ns prior to the peak power. According to our benchmarked 2D radiative hydrodynamic simulation (see Methods section), a peak areal density ($\rho R$) of ~0.06 g cm⁻² can be achieved at 2.6 ns (Fig. 1b) under these experimental configurations. In joint shots (see Table 1 for detailed experimental parameters), the short pulse (1.5 ps in FWHM) LFEX laser was injected in the equatorial plane at different times around the peak compression (2.6 ns), generating a forward moving fast electron beam. It was focused 230 μm ahead of the target center (Fig. 1a), which corresponded to ~0.5 $n_c$ ($n_c = 10^{21}$ cm⁻³ for the LFEX laser) as predicted by the simulation. The generated fast electrons transported through the compressed target, colliding with the Cu atoms and exciting Kα emission. The spatial distribution of the Cu Kα emission was recorded by a narrow-band (5 eV) Cu-Kα imager; while the total Cu Kα photon number was measured by a calibrated planar highly oriented pyrolytic graphite (HOPG) crystal spectrometer. Fast electrons escaping from the target were measured by electron spectrometers (ESMs) at different directions. An x-ray streak camera (XSC) served to detect the temporal and spatial evolutions of the target self-emission (the thermal emission in photon energy range

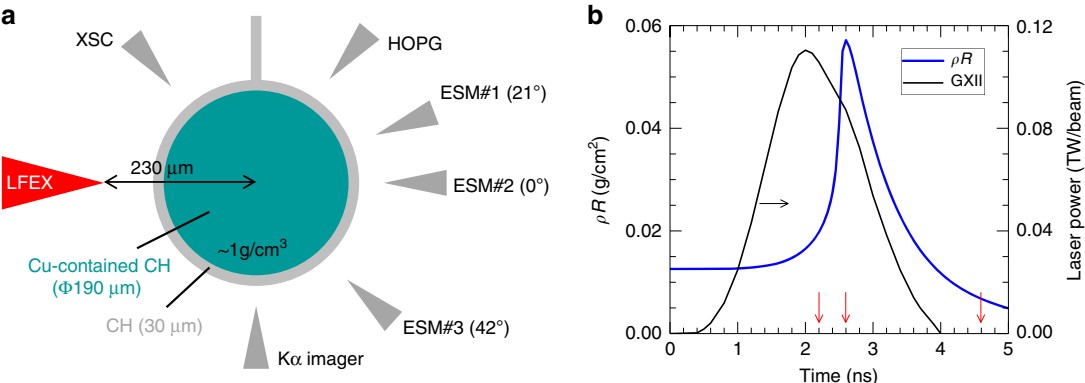

**Fig. 1 Experimental setup, laser power and areal density. a** Target parameters, the LFEX laser and detectors. A Cu-Kα imager and a HOPG crystal spectrometer are used to detect the 2D image and the spectrum of the Cu Kα emission, respectively. Three electron spectrometers (ESMs) are used to measure the energy spectra of the escaping fast electrons at different directions. An X-ray streak camera (XSC) functions to detect the target self-emission. **b** Experimentally measured beam power of GXII lasers (black) and simulated target areal density (blue) as a function of time. The LFEX injection times (2.2 ns, 2.6 ns, and 4.6 ns) in joint shots are marked by red arrows.

**Table 1 Experimental parameters of the joint shots.**

| Shot ID | $t_{LFEX}$ (ns) | $E_{LFEX}$ (J) | $I_{LFEX}$ (W cm$^{-2}$) | $\gamma$ | $\rho R$ (mg cm$^{-2}$) | $N_{K\alpha}$ (ph sr$^{-1}$) |
|---|---|---|---|---|---|---|
| #41261 | 2.2 | 396 | $3.4 \times 10^{18}$ | 1.9 | 19.7 | $6.2 \times 10^{10}$ |
| #41266 | 2.6 | 423 | $3.7 \times 10^{18}$ | 1.9 | 57.2 | $2.3 \times 10^{10}$ |
| #41255 | 4.6 | 191 | $1.7 \times 10^{18}$ | 1.5 | 6.8 | $1.7 \times 10^{11}$ |

The injection time ($t_{LFEX}$), energy ($E_{LFEX}$), peak intensity ($I_{LFEX}$), and Lorentz factor ($\gamma$) of the LFEX laser, and the fast-electron-produced Cu Kα photons ($N_{K\alpha}$) in each of the three joint shots (#41261, #41266, and #41255) were experimental measurements. The corresponding target areal densities ($\rho R$) were simulated results

of 1–10 keV), based on which the 2D radiative hydrodynamic code (FLASH, ref. [32]) was benchmarked.

A solid sphere rather than a conventional shell was used because of two reasons. First, a mass-equivalent solid sphere is more hydrodynamically stable than a shell during the compression under the current conditions of the GXII lasers[33,34], thus promising a denser core at the peak compression with a very good shot-to-shot reproducibility. Second, the core temperature at the peak compression in a sphere target is lower than that in a shell target, facilitating the detection of fast electron transport by the stimulated Cu Kα emission, because the high temperature could shift the Cu Kα line outside the narrow bandwidth of the Kα imager, leading to signal diminishing in the core region[16,35].

Apart from the joint shots, where both the GXII lasers and the LFEX laser were applied, some GXII-only shots were also performed by switching off the LFEX laser. In the GXII-only shot, the Kα imager measured the target self-emission in the detectable photon energy range, as well as the Cu Kα emission stimulated by the GXII-produced suprathermal electrons; while in the joint shot, additional Cu Kα emission due to the LFEX-produced fast electrons was observed. By comparing the results between the GXII-only and joint shots, the fast electron induced Cu Kα images were obtained. Applying the same method to the HOPG results, one obtained the total Cu Kα photon numbers produced by the fast electrons.

**Experimental results**. Figure 2a displays a typical target self-emission recorded by the XSC in experiments, which indicates compression of the target surface up to 3 ns. This compression process is well reproduced by the simulated target self-emission obtained from the benchmarked FLASH code, as shown in Fig. 2b, c. This good agreement between the simulation and the

experiment validates the reliability of the FLASH code in predicting the hydrodynamic behavior of the target. The predicted density and temperature profiles at the injection times of the LFEX laser are shown in Fig. 3, while the predicted history of the areal density is shown in Fig. 1b. These results indicate that, because of the laser ablation, a shock is launched in the target, which moves inward until 2.6 ns, forming a high-density shell surrounding the target center (see Fig. 3a). At 2.6 ns, the target reaches its peak compression, resulting in a core with a radius of ~70 μm. The areal density and the density at the target center at this moment correspond to 0.057 g cm$^{-2}$ and 20 g cm$^{-3}$, respectively. After 2.6 ns, the target starts to decompress, reaching a size of ~120 μm in half width at half maximum at 4.6 ns. The laser ablation produces a hot (~2 keV) and large density gradient scalelength (~60 μm at the critical density surface for the LFEX laser, see Supplementary Fig. 1 and Supplementary Discussion for more details) coronal plasma around the time of peak compression (e.g., 2.2 ns and 2.6 ns). While at 4.6 ns, the coronal temperature drops below 50 eV due to the termination of the GXII lasers; the density gradient scalelength decreases to ~15 μm due to the decompression of the target.

The fast electron induced Cu Kα images at different times are shown in Fig. 4a–c, while the corresponding Cu Kα photon numbers measured by the HOPG spectrometer are listed in Table 1. Figure 4a–c reflects directly the fast electron transport in a compressed target at different evolution phases. The size of the emission region decreases from 2.2 ns to 2.6 ns and then increases thereafter, corresponding, respectively, to the compressing (from Fig. 4a, b) and decompressing (from Fig. 4b, c) phases of the target.

At 2.2 ns, 0.4 ns before the peak compression, most of the Cu Kα emission is produced in the inward-moving shocked region

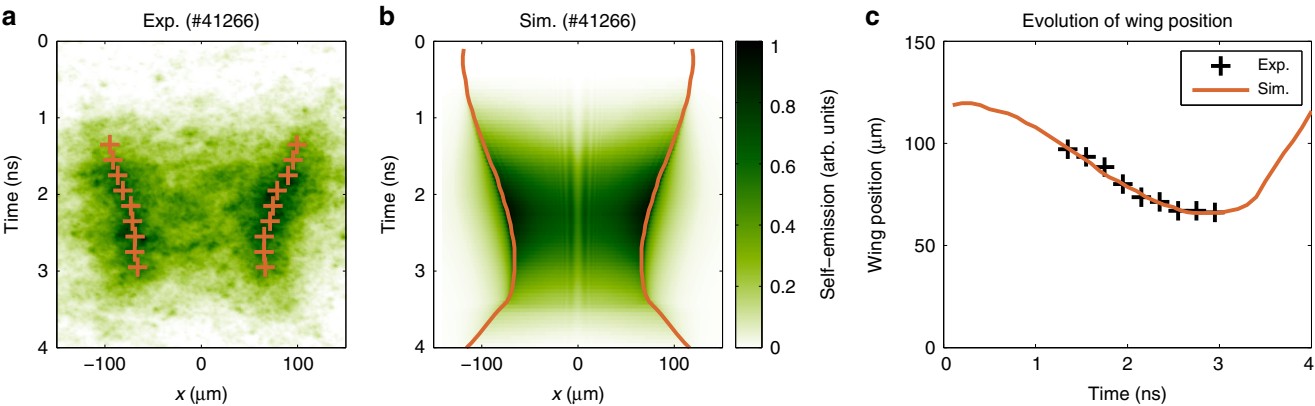

**Fig. 2 Temporal and spatial evolution of the target self-emission. a** Experimentally measured and **b** simulated target self-emission for a typical shot (#41266). The positions of the wings, marked with crosses and lines in **a**, **b**, respectively, are plotted in **c** as a function of time. Here, the positions of the wings are defined as the positions of the peak emission at each side of the target.

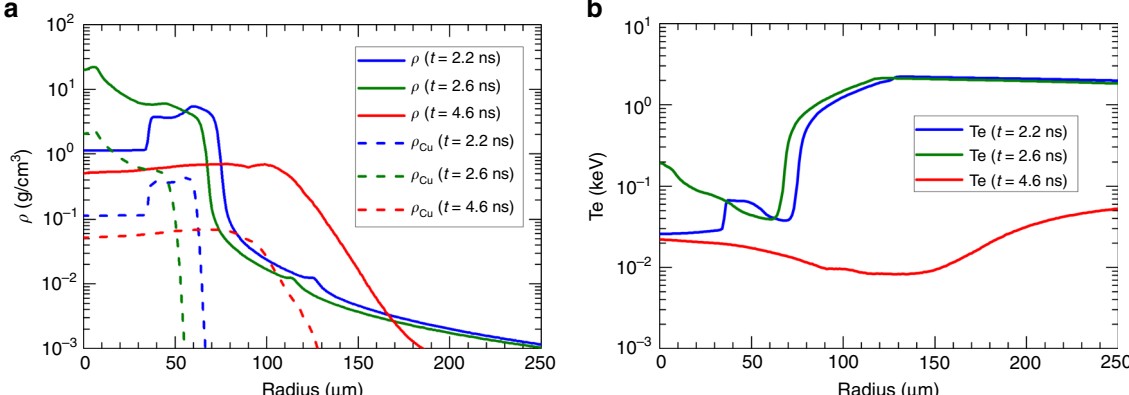

**Fig. 3 Simulated density and temperature profiles at different times.** Radial distributions of **a** mass densities of both the whole target (solid) and the Cu atoms (dashed) and **b** electron temperatures from FLASH simulations at different times: 2.2 ns (blue), 2.6 ns (green), and 4.6 ns (red). Since the inner Cu contained sphere is coated by a CH layer, the size of the Cu contained region is always smaller than the target size, being ~60 μm (2.2 ns), ~50 μm (2.6 ns), and ~100 μm (4.6 ns) in radius, respectively. The density gradient scalelengths at the critical density surface (r ~ 160 μm) for the short pulse LFEX laser are, respectively, ~60 μm (2.2 ns), ~60 μm (2.6 ns), and ~15 μm (4.6 ns).

due to its much higher density of Cu atoms than the rest region, as shown in Fig. 3a. The asymmetry (stronger emission on the upstream (left) side) shown in Fig. 4a is caused by the injection of the fast electrons from the LFEX entrance side. Due to the intrinsic angular divergence and energy deposition, the density of fast electrons decreases as they propagate from left to right, so that the Cu Kα intensity decreases accordingly.

At 2.6 ns, a small emission region with a side-to-side diameter of ~100 μm is observed (see Fig. 4b), which agrees well with the diameter (100 μm) of the Cu contained region at the time of peak compression predicted by the FLASH simulation (see Fig. 3a). The weak emission at this time could be explained possibly by three factors. First, the core subtends a small solid angle (~0.2π) from the fast electron source, leading to a reduction in the population of fast electrons that can hit the core. Second, the large target opacity due to the high areal density results in an intensity decay of the Cu Kα emission. Third, the enhancement of the core temperature (see Fig. 3b) shifts and broadens the Cu Kα line, reducing the detection efficiency of the narrow-band Kα imager[35]. It is worth noting that this is the direct observation of the fast electron transport through a high-density core. With this result and the Cu Kα photon number given by the HOPG

spectrometer, the energy deposited in the core by the fast electrons can be estimated, which will be discussed later.

At 4.6 ns, the target has decompressed to a large size (see Fig. 3a) and a water-drop-like Cu Kα image is observed (see Fig. 4c), which clearly reveals the angular divergence of the fast electrons. By taking advantage of this spatial feature in the Cu Kα image, the divergence angle of the fast electrons can be estimated. In contrast, at 2.2 ns and 2.6 ns, the Cu contained regions are so small that the shapes of the Cu Kα images are primarily dependent on the target geometry rather than the spatial distribution of fast electrons, making it impossible to estimate the divergence angle of fast electrons at these times.

**Laser-to-core energy coupling efficiency.** The energy deposited by fast electrons is inferred by taking advantage of the fact that the fast electrons deposit their energy and excite Cu Kα emission concomitantly when passing through the Cu contained plasma[16,19]. In principle, as long as the plasma conditions and the cross-section of Cu Kα emission are known, the measured Cu Kα emission, including its absolute intensity and spatial features, allows one to derive the information of the fast electron beam, which is usually characterized by an exponential distribution for

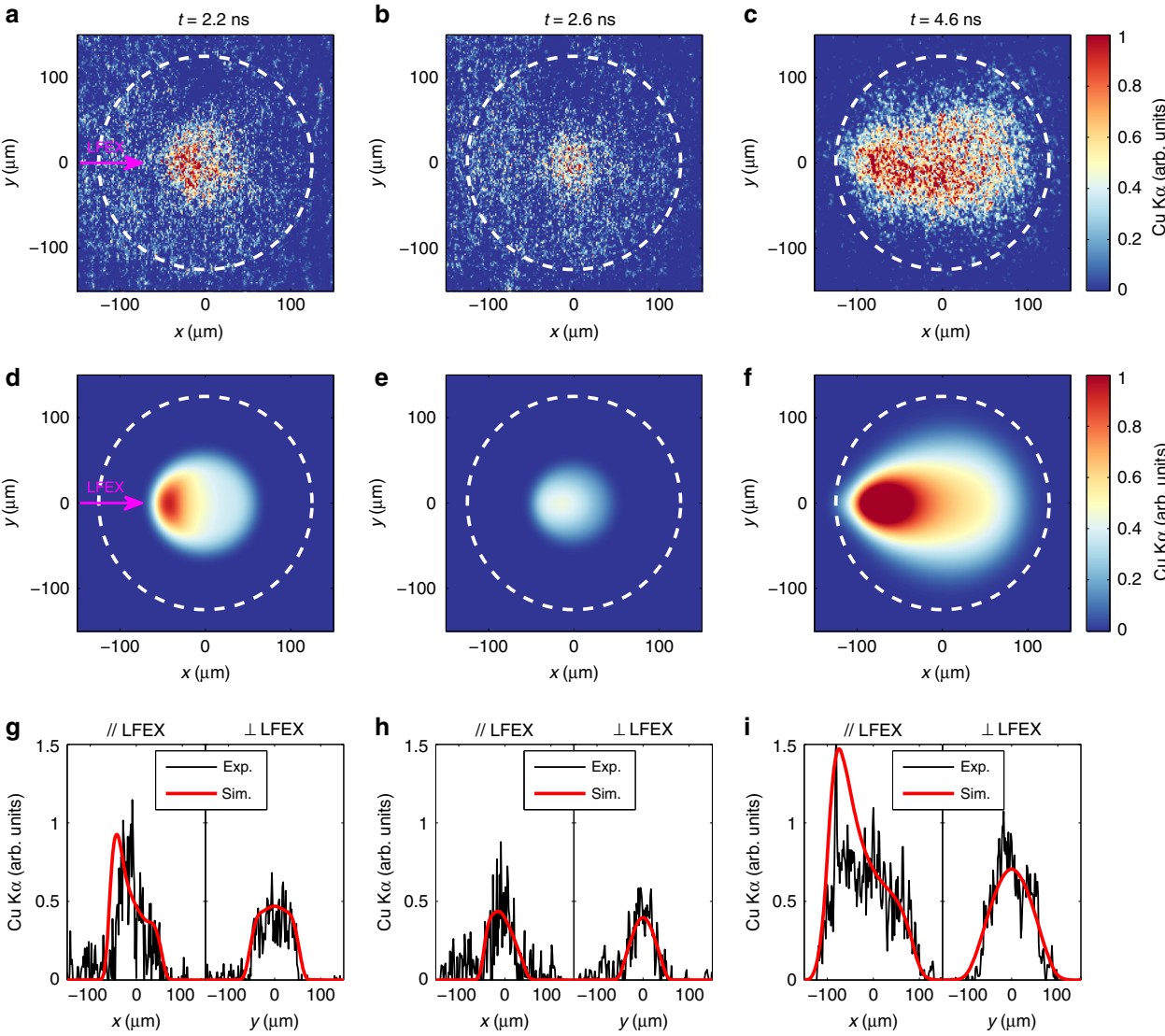

**Fig. 4 Cu Kα images induced by fast electrons as a function of LFEX injection time. a–c** Fast electron induced Cu Kα images measured in experiments when the LFEX laser is injected at 2.2 ns, 2.6 ns, and 4.6 ns, respectively. The background attributable to the drive lasers, which is measured in the GXII-only shot (see Supplementary Fig. 2), has been subtracted. **d–f** Simulated Cu Kα images induced by fast electrons at the corresponding LFEX injection times. The details of the simulation are described in the text. In both measured and simulated images, the LFEX laser is injected from the left side, as marked in **a** and **d**. The white dashed circles represent the original size of the target. **g–i** Horizontal (left) and vertical (right) lineouts across the target center from experiments (black) and simulations (red) at the corresponding times, respectively.

the energy spectrum and by a Gaussian distribution for the angular divergence. With the derived information, the deposited energy can be estimated based on the collisional stopping power. Following this idea, a code named eTrans in 2D cylindrical geometry has been developed to simulate the transport and energy deposition of a fast electron beam, as well as its excitation of the Cu Kα emission in a given plasma (see Methods section).

In our experiments, the short pulse laser had to pass through a long density scalelength coronal plasma before reaching the critical density surface (see Fig. 3a); as a result, an additional high-temperature component in the fast electrons would be produced in the underdense region, as observed in both experiments[36] and simulations[37]. Therefore, in our eTrans simulations, the input fast electron beam consisted of two components with different slope temperatures ($T_1$ and $T_2$, with $T_1 < T_2$). Each component was further parameterized by its own divergence angle ($\theta_{1,2}$, in FWHM) and population ($N_{1,2}$). Because

the $T_2$-component was energetic enough to escape from the target and be detected by the ESMs, its parameters ($T_2$, $\theta_2$, and $N_2$) were obtained by fitting the experimental data, as listed in Table 2. The estimated total energy of this $T_2$-component ($E_2$) was just several joules, corresponding to a few percentages of the LFEX laser energy ($\eta_{L \to T_2}$). As a result, the proportion ($\eta_{K\alpha}$) of the Cu Kα photons excited by this $T_2$-component was less than 1% in the experimentally measurements. It meant that the Cu Kα photons measured in experiments and hence the deposited energy resulted dominantly from the $T_1$-component. Theoretically, $T_1$, $\theta_1$, and $N_1$ could be obtained by fitting the measured Cu Kα emission. However, the small size of the Cu Kα emission in our experiments (see Fig. 4) revealed limited spatial features, making it difficult to derive $T_1$ and $\theta_1$. All we could obtain were simply the upper limit of $T_1$ and the lower limit of $\theta_1$ (see Supplementary Fig. 5 and Supplementary Discussion for more details). Therefore, $T_1$ and $\theta_1$ were chosen as free parameters in the eTrans simulations, while

$N_1$ was determined by matching the simulated Cu Kα photons with the experimental data. Fortunately, as will be shown below, within a large parameter domain of interest (e.g., $0.5\,\mathrm{MeV} \leq T_1 \leq 3\,\mathrm{MeV}$ and $\theta_1 \geq 30°$ for the $t_{\mathrm{LFEX}} = 2.6\,\mathrm{ns}$ case), the simulated Cu Kα emission and hence the deposited energy depended strongly on $N_1$ while weakly on $T_1$ and $\theta_1$, which therefore allowed us to estimate the deposited energy even with lack of exact $T_1$ or $\theta_1$.

Examples of the simulated Cu Kα images are shown in Fig. 4d–f, with their horizontal and vertical lineouts plotted in Fig. 4g–i together with the corresponding experimental data. In these simulations, $T_1 = 1\,\mathrm{MeV}$ and $\theta_1 = 40°$ were used for the $T_1$-component fast electrons, while the plasma parameters were from the FLASH simulations. It indicated that the Cu Kα images were weakly dependent on $T_1$, as long as $T_1 > 0.1\,\mathrm{MeV}$. For 2.2 ns and 2.6 ns, due to the small size of the Cu contained region, these images were also weakly dependent on $\theta_1$, as long as $\theta_1 > 20°$. While for 4.6 ns, the comparison between experimental data and simulation results indicated that $\theta_1 = 30°$–$40°$ (see Supplementary Fig. 4 and Supplementary Discussion). These simulated Cu Kα images well reproduced the experimental data, including the sizes of the emission region, the relative intensities at different times, and features such as the asymmetry at 2.2 ns and the water-drop-like shape at 4.6 ns. This good agreement in the Cu Kα emission between simulation and experiment once again justified the reliability of the FLASH simulation. At 2.6 ns, the simulation indicated that the target opacity and the detection efficiency of the Kα imager did play important roles in the weak Cu Kα

emission, leading to intensity reduction by factors of 27% and 23%, respectively, at the image center.

At the time of peak compression (2.6 ns), a series of eTrans simulations were performed by scanning $T_1$ and $\theta_1$ in the domain of interest ($0.5\,\mathrm{MeV} \leq T_1 \leq 3\,\mathrm{MeV}$ and $\theta_1 \geq 30°$, see Supplementary Discussion) to estimate the laser-to-core energy coupling efficiency ($\eta_{\mathrm{L}\to\mathrm{core}}$). Here, $\eta_{\mathrm{L}\to\mathrm{core}}$ is defined as the ratio of the energy deposited by fast electrons inside the core (70 μm in radius, see Fig. 3a) to that of the LFEX laser. For each pair of $T_1$ and $\theta_1$, the population ($N_1$) and hence the total energy ($E_1$) of $T_1$-component fast electrons were obtained, which allowed the calculation of the laser-to-electron energy conversion efficiency ($\eta_{\mathrm{L}\to\mathrm{e}} = (E_1 + E_2)/E_{\mathrm{LFEX}}$), as summarized in Fig. 5a. In most of the cases, $\eta_{\mathrm{L}\to\mathrm{e}}$ was physically reasonable, at the level of $\lesssim 30\%$. While in some cases with $\theta_1 \geq 90°$, unphysically large values (e.g., $\geq 50\%$) of $\eta_{\mathrm{L}\to\mathrm{e}}$ were derived, indicating that either $T_1$ or $\theta_1$ was overestimated in these cases. We note that this overestimation of $T_1$ or $\theta_1$ did not affect the estimation of $\eta_{\mathrm{L}\to\mathrm{core}}$. As shown in Fig. 5b, $\eta_{\mathrm{L}\to\mathrm{core}}$ depended weakly on $T_1$ and $\theta_1$ when $T_1 > 1\,\mathrm{MeV}$ and $\theta_1 > 60°$, respectively. In the parameter domain of interest ($0.5\,\mathrm{MeV} \leq T_1 \leq 3\,\mathrm{MeV}$ and $\theta_1 \geq 30°$), $\eta_{\mathrm{L}\to\mathrm{core}} = (0.83 \pm 0.23)\%$ was obtained, which corresponded to an energy of $3.5 \pm 1.0\,\mathrm{J}$ deposited inside the core. Note that our eTrans simulations were constrained by the total Cu Kα photon number, which had an accuracy of ±30% in measurement. By taking this additional uncertainty into account (added in quadrature), $\eta_{\mathrm{L}\to\mathrm{core}} = (0.8 \pm 0.3)\%$.

### Table 2 Parameters of $T_2$-component fast electrons.

| $t_{\mathrm{LFEX}}$ (ns) | $T_2$ (MeV) | $\theta_2$ (°) | $N_2$ (sr$^{-1}$) | $E_2$ (J) | $\eta_{\mathrm{L}\to T_2}$ | $\eta_{\mathrm{K}\alpha}$ |
|---|---|---|---|---|---|---|
| 2.2 | 13.9 | 33.2 | $1.1 \times 10^{13}$ | 8.7 | 2.2% | 0.4% |
| 2.6 | 15.7 | 30.7 | $4.3 \times 10^{12}$ | 3.5 | 0.8% | 1.0% |
| 4.6 | 8.0 | 33.0 | $6.3 \times 10^{12}$ | 3.0 | 1.6% | 0.1% |

The slope temperature ($T_2$), divergence angle ($\theta_2$), and population ($N_2$) of $T_2$-component fast electrons at different LFEX injection times ($t_{\mathrm{LFEX}}$) were obtained by fitting the ESM data (see Supplementary Fig. 3 and Supplementary Discussion for more details). These parameters allowed the derivation of the energies ($E_2$) of these $T_2$-component fast electrons and hence their proportions ($\eta_{\mathrm{L}\to T_2}$) in the LFEX laser energies. Simulations with the eTrans code were performed to estimate the Cu Kα photons excited by these $T_2$-component fast electrons, from which their proportions ($\eta_{\mathrm{K}\alpha}$) in the experimentally measured results were obtained

## Discussion

We figured out that the laser-to-core energy coupling efficiency could be improved by optimizing the implosion and super-penetration performances, because it was limited by two aspects in experiments reported here. One aspect was the low areal density ($\rho R \sim 0.06\,\mathrm{g\,cm^{-2}}$) of the core, which resulted from the low driver energy ($\sim 200\,\mathrm{J\,beam^{-1}}$) and the target preheat caused by the driver-produced suprathermal electrons[38]. The kinetic energy of fast electrons optimal for heating a core of $0.06\,\mathrm{g\,cm^{-2}}$ ranges from 200 keV to 500 keV. When electrons with a slope temperature of ~1 MeV were transmitting through a core of this areal density, only ~10% of the energy could be deposited[16]; in contrast, 40–50% of the energy could be deposited for an areal

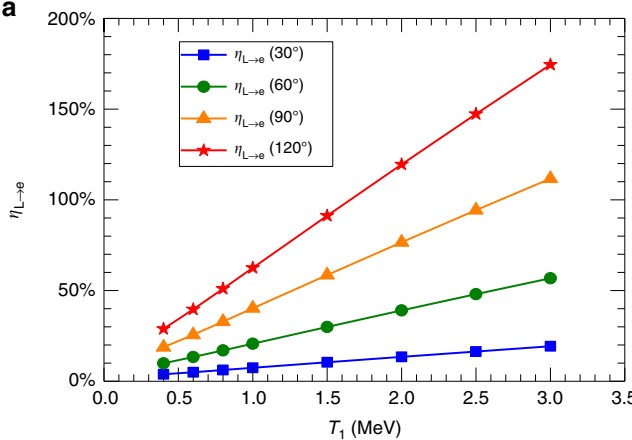

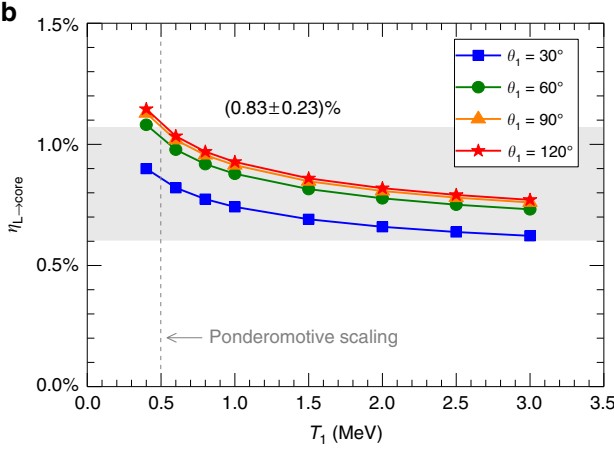

**Fig. 5 Simulated results for the $t_{\mathrm{LFEX}} = 2.6$ ns case.** Simulated (**a**) laser-to-electron energy conversion efficiency ($\eta_{\mathrm{L}\to\mathrm{e}}$) and (**b**) laser-to-core energy coupling efficiency ($\eta_{\mathrm{L}\to\mathrm{core}}$) as a function of the slope temperature ($T_1$) and the divergence angle ($\theta_1$) of the $T_1$-component fast electrons. These simulations are constrained by matching the simulated Cu Kα photons with the experimental measurement. $\eta_{\mathrm{L}\to\mathrm{e}}$ increases rapidly as $T_1$ and $\theta_1$ increase. $\eta_{\mathrm{L}\to\mathrm{core}}$ depends weakly on $T_1$ and $\theta_1$ when $T_1 > 1\,\mathrm{MeV}$ and $\theta_1 > 60°$, respectively. For the LFEX intensity in the experiments, $\sim 3.7 \times 10^{18}\,\mathrm{W\,cm^{-2}}$, the slope temperature given by the ponderomotive scaling[27] is ~0.5 MeV, as marked by the dashed line. In the domain of $[0.5\,\mathrm{MeV}, 3.0\,\mathrm{MeV}] \times [30°, +\infty)$, $\eta_{\mathrm{L}\to\mathrm{core}} = (0.83 \pm 0.23)\%$.

density of 1.3 g cm$^{-2}$ (ignition level attainable in the NIF-scale, corresponding to an optimal electron energy ranging from 2 MeV to 5 MeV). The other aspect was the low energy carried by the fast electrons to the core region. Taking $T_1 = 1$ MeV as an example, our eTrans simulations indicated that only 26–39 J energy was transported to the core region, corresponding to 6–9% of the LFEX energy (423 J), as shown in Fig. 6. This low energy could be limited by three factors. (i) The laser-to-electron energy conversion efficiency ($\eta_{L \to e}$) might be very low (e.g., $\eta_{L \to e} = 10\%$ at $\theta_1 = 40°$, dashed lines in Fig. 6), which could be a result of spending too much energy in forming the plasma channel or creating laser and electron filaments[30], leaving little energy to produce fast electrons. (ii) The fast electrons might have a very large

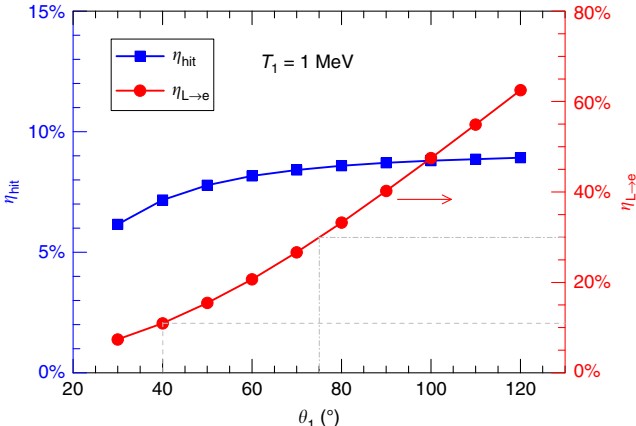

**Fig. 6 Energy carried by fast electrons for the $t_{LFEX} = 2.6$ ns case.** The fraction of the LFEX energy ($\eta_{hit}$, blue squares, left axis) carried by fast electrons that are capable of hitting the core and laser-to-electron energy conversion efficiency ($\eta_{L \to e}$, red dots, right axis) are plotted as a function of the divergence angle ($\theta_1$) of the $T_1$-component fast electrons. In these simulations, $T_1 = 1$ MeV and the calculated Cu Kα photons are set to match the experimental measurement. Only 6–9% of the LFEX energy is transported to the core region, indicating a low $\eta_{L \to e}$ (as marked by the dashed lines), or a large $\theta_1$ (as marked by the dash-dotted lines) in the fast electrons, or a large distance from the fast electron source to the core.

divergence angle (e.g., $\theta_1 = 75°$ at $\eta_{L \to e} = 30\%$, dash-dotted lines in Fig. 6), which could be caused by beam filamentation, laser hosing and channel bifurcation[39] when the LFEX laser was propagating through the long scalelength coronal plasmas. (iii) The fast electron source might be far away from the core. Previous experiments[29,30] showed that the plasma channel created by a 10 ps high-intensity ($\sim 10^{19}$ W cm$^{-2}$) laser stopped in the underdense region ($0.6 n_c$), probably due to beam filamentation and plasma pileup in front of the channel.

To identify the dominant factors responsible for the low energy carried to the core region, 2D PIC simulations (see Methods section) were performed for LFEX injection times at 2.6 ns and 4.6 ns. In these simulations, the LFEX parameters were from the experiments while the plasma conditions were from the FLASH simulations. The simulated results (see Supplementary Figs. 6–9 for more details) showed that strong laser filamentation and channel bifurcation took place in the $t_{LFEX} = 2.6$ ns case, where the coronal plasma had a long density scalelength ($\sim 60$ μm); as a result, the plasma channel stopped in the underdense region and the generated fast electrons were widely distributed in the transverse direction, as shown in Fig. 7. Because of the long density scalelength in this case, pulse depletion also played an important role in the reduced channel depth[33]. While for the $t_{LFEX} = 4.6$ ns case, where the density scalelength in the coronal region is much shorter ($\sim 15$ μm), the plasma channel arrived at the relativistic critical density ($\gamma n_c$, where $\gamma$ is listed in Table 1) surface, and a much narrower electron beam was produced (see Fig. 7b, c). This narrow electron beam was supported by the experimentally inferred divergence angle of 30°–40° in FWHM (see Supplementary Discussion), indicating that a well collimated fast electron beam could be produced by interacting a high intensity laser with a short density scalelength plasma. These results demonstrated that, rather than the low $\eta_{L \to e}$, the large divergence angle and long distance from the core of the fast electrons caused by the strong laser filamentation and channel bifurcation (see Fig. 7a) were the dominant reasons for the low energy transported to the core region at the time of peak compression (2.6 ns).

These findings uncover the importance of applying an additional channeling pulse in this super-penetration scheme, because the plasma channel preformed by the channeling pulse not only

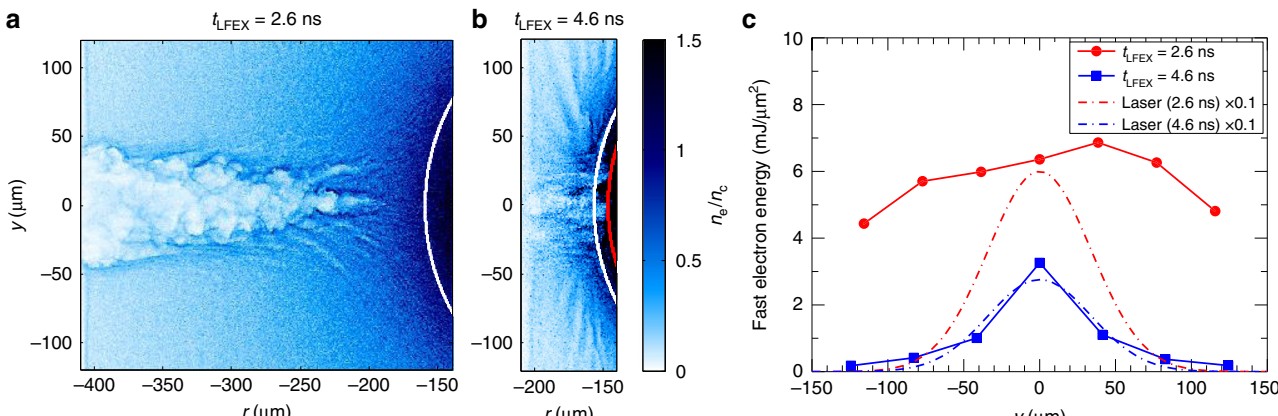

**Fig. 7 Two-dimensional particle-in-cell simulation results. a, b** Electron density profiles obtained 0.2 ps after the LFEX laser terminates are shown in **a** and **b** for cases of $t_{LFEX} = 2.6$ ns and $t_{LFEX} = 4.6$ ns, respectively. The horizontal axis denotes the distance from the target center. The white and red curves represent the classic critical density ($n_c$) and relativistic critical density ($\gamma n_c$) surfaces, respectively, at the injection time of the LFEX laser. **c** Transverse distributions of time-integrated fast electron energies measured at $r \sim -140$ μm (1 μm beneath the rear surface of the plasma). These measurements only take into account the electrons with energy above 0.6 MeV. The red dots and blue squares are for the $t_{LFEX} = 2.6$ ns and $t_{LFEX} = 4.6$ ns cases, respectively. Their transverse distributions of the time-integrated laser energies multiplied by a factor of 0.1 are, respectively, plotted by red and blue dash-dotted lines as reference.

helps to suppress the filamentation of the second heating pulse but also provides a short density scalelength plasma at the channel front. As demonstrated experimentally, by optimizing the experimental configuration, a plasma channel up to critical density surface can be created[29,40]. Besides, a magnetic field could be produced by the channeling pulse, collimating the fast electrons generated by the second pulse[41]. Therefore, a high laser-to-core energy coupling efficiency could be expected in this optimized double-pulse scheme. Given that the experimentally inferred laser-to-electron conversion efficiency is above 30% (ref. [42]), if the fast electrons are well guided to the core region, a laser-to-core coupling efficiency of 3% could be expected for the 0.06 g cm$^{-2}$ core plasma on the GEKKO-LFEX facility. While for the NIF-scale core plasma (1.3 g cm$^{-2}$ in areal density), 40–50% of the energy of fast electrons could be deposited when they pass through the core[16]; therefore, a laser-to-core energy coupling efficiency of as high as $\gtrsim$12% could be expected, which is comparable to that expected for the cone-in-shell scheme[16]. However, it is worth noting that, compared to the cone-in-shell scheme, this super-penetration scheme has a simpler experimental configuration and requires no external assistance for electron collimation, which are good advantages for the ignition experiment. Also due to the absence of the cone, multiple short-pulse beams incident from different directions (e.g., six orthogonal directions) can be applied in super-penetration scheme. Compared to the cone-in-shell scheme, this configuration either promises more energy coupled to the core by increasing the short-pulse beams or relaxes the engineering requirement of optical elements by reducing the intensity of each beam.

## Methods

**Target.** The target used in the experiments consisted of two parts: an inner solid Cu contained CH sphere and an outer CH coating layer. The inner sphere with a diameter of 190 μm was made of Cu(II) oleate [Cu(C$_{17}$H$_{33}$COO)$_2$][31], resulting in a Cu concentration of ~10% in weight; while the outer coating layer with a thickness of 30 μm was made of parylene. The densities in both parts were ~1 g cm$^{-3}$. Our previous experiments in similar configurations indicated that, when a ~20 μm thick coating layer was used, plenty of suprathermal electrons produced by the drive (GXII) lasers were able to penetrate into the inner region, resulting in strong background Cu Kα emission[38]. Therefore, a thicker (30 μm) coating layer was used in the experiments reported here to reduce the background Cu Kα emission, thereby facilitating the detection of fast electron transport.

According to the radiative hydrodynamic simulations, the peak areal density in this solid sphere target is ~0.06 g cm$^{-2}$, higher than that in a mass-equivalent shell target (~0.04 g cm$^{-2}$); while the core temperature in the former is $\lesssim$200 eV, colder than that in the later (~380 eV). With the temperature dependence of detection efficiency of the Cu Kα imager taken into account, this solid sphere target is more suitable for detecting the fast electron transport than the conventional shell target.

**Lasers.** Twelve GEKKO lasers were applied to compress the target, with each laser operating at a wavelength of 526 nm (2ω) with an energy of ~200 J. All of the lasers were focused with an F/3 lens, resulting in a focal spot of ~170 μm in full width at half maximum (FWHM) at the target surface. The pulse shape was Gaussian with a FWHM of 1.6 ns, leading to a peak intensity of ~4.8 × 10$^{14}$ W cm$^{-2}$. The pulse shape, as shown in Fig. 1b, was measured by an oscilloscope in each shot. In joint shots, the short pulse LFEX laser was injected into the target in the equatorial plane to produce a fast electron beam. Its wavelength and pulse duration were, respectively, 1053 nm (1ω) and 1.5 ps in FWHM. It was focused 230 μm ahead of the target center [see Fig. 1a] with a focal spot of 80 μm in FWHM. With energy ranging from ~190 J to ~420 J, a peak on-target intensity of ~3.7 × 10$^{18}$ W cm$^{-2}$ could be achieved, as shown in Table 1. The relative timing between the GEKKO and LFEX lasers was measured offline in two successive shots by, respectively, irradiating the GEKKO and LFEX lasers at very low energy onto a diffuser placed at the target chamber center. The scattered light was detected by a high-speed photodiode coupled with an oscilloscope. In both shots, the oscilloscope was triggered by the GEKKO laser; consequently, the on-target delay of LFEX relative to the peak of GEKKO was obtained by comparing the recorded signals in these two shots. The delay measured in this way had an uncertainty of ±150 ps due to the intrinsic jitter of the LFEX timing.

**Diagnostics.** The Cu Kα imager used a spherically bent quartz (2131) crystal to image the emission within a spectral bandwidth of 5 eV (FWHM) centered at 8.048 keV (Cu Kα1) from the target onto an image plate. The magnification and spatial resolution were, respectively, 17.5 and 13 μm (FWHM). A 10 μm Cu filter was placed in front of the image plate to reduce the background. This imager was installed perpendicularly to the LFEX axis.

A planar highly oriented pyrolytic graphite (HOPG) crystal spectrometer, also perpendicular to the LFEX axis, was applied to measure the x-ray spectrum in the range of 7.7–9.2 keV. It was absolutely calibrated for the Cu Kα line emission in a separate experiment by comparing with the results measured by a calibrated single-photon counting camera, resulting in a calibration factor with an accuracy of ±30%. Because of its large detectable energy range, the HOPG spectrometer would not be affected by the shifting and broadening of the Cu Kα line due to temperature enhancement.

**Simulations.** The compression of the target was simulated by the FLASH code[32] in 2D-cylindrical geometry coupled with a hot-electron-preheat module[38]. FLASH is a radiative hydrodynamic code using 3D-in-2D ray trace algorithm for the laser energy deposition. Therefore, the twelve GXII lasers could be set up exactly the same as the experimental conditions, including the beam energy, pulse shape, incident direction, F-number and spot size. The radiation transport was solved by a multigroup (6 groups) radiation diffusion model with tabulated opacity. A flux-limited model with the Larsen square-root flux limiter was applied to the calculation of thermal conduction. The hot-electron-preheat module was coupled to take into account the effect on the target compression caused by the energy deposition of GXII-produced suprathermal electrons. The necessity of considering this preheat effect was demonstrated by our previous experiments under similar configurations[38]. This simulation code was benchmarked by comparing the simulated temporal and spatial evolutions of the target self-emission (i.e., the XSC result) with the experimental data, as shown in Fig. 2. Details of this simulations was also introduced in ref. [38].

A 2D cylindrically symmetric code, eTrans, has been developed to simulate the transport of an electron beam in a specific plasma. In eTrans, the electrons propagate straightly along their initial velocity directions and their transport is governed by the collisional stopping power with scattering effect considered[43]; namely, fields are neglected. With this stopping power, the energy deposited in the plasma is calculated. With the Cu Kα emission cross-section[44] and the spatial distribution of the Cu atoms, eTrans is also able to calculate the Cu Kα emission along the path of fast electrons. By taking into account the 3D projection along the line of sight of the Kα imager, the target opacity, and the detection efficiency of the Kα imager due to temperature enhancement, a 2D Cu Kα image is obtained. eTrans also provides the total (spatially integrated) Cu Kα photon number (with the opacity considered but the detection efficiency neglected), so that the result can be compared with the experimental data given by a temperature-independent detector, such as the HOPG spectrometer.

In our eTrans simulations, the plasma conditions were given by the FLASH output, while the fast electron beam consisted of two components, with each characterized by an exponential distribution for the energy spectrum and by a Gaussian distribution for the angular divergence:

$$f(E,\theta) = \frac{N_1}{T_1}\exp\left(-\frac{E}{T_1}\right)\exp\left(-4\ln2\frac{\theta^2}{\theta_1^2}\right) + \frac{N_2}{T_2}\exp\left(-\frac{E}{T_2}\right)\exp\left(-4\ln2\frac{\theta^2}{\theta_2^2}\right),$$

(1)

where $T_1 < T_2$. The $T_2$-component was completely determined by the experimental data from ESMs (see Supplementary Fig. 3). For the $T_1$-component, because of its low energy, it suffers strong deposition and deflection when passing through the target; as a result, it could hardly be measured by the ESMs. Therefore, the slope temperature ($T_1$) and the divergence angle in FWHM ($\theta_1$) were chosen to be free parameters, while the population ($N_1$) was always constrained to match the calculated total Cu Kα photons with the experimental data measured by the HOPG spectrometer. All these fast electrons were injected into the target at the critical density surface ($n_c = 10^{21}$ cm$^{-3}$).

The 2D particle-in-cell (PIC) code, FISCOF (ref. [45]), was used to simulate the propagation of the short pulse LFEX laser in the coronal plasmas and the production of fast electrons. The parameters characterizing the LFEX laser were obtained from the experiments. Namely, both spatial and temporal profiles were Gaussian functions with FWHMs of 80 μm and 1.5 ps, respectively. For the injection time at 2.6 ns (4.6 ns), the experimentally measured LFEX energy was ~420 J (~190 J), resulting in a peak intensity of ~3.7 × 10$^{18}$ W cm$^{-2}$ (~1.7 × 10$^{18}$ W cm$^{-2}$). The density profile of the fully ionized CH plasma was taken from the FLASH-simulated results at the corresponding time, as shown in Supplementary Fig. 1a, ranging from 0.1$n_c$ to 1.5$n_c$ (3.9$n_c$) for $t_{LFEX}$ = 2.6 ns ($t_{LFEX}$ = 4.6 ns). The maximum density was chosen to avoid penetrating of the LFEX laser through the plasma. As a result, the simulation box was 240 × 280 μm$^2$ (140 × 400 μm$^2$) for $t_{LFEX}$ = 2.6 ns ($t_{LFEX}$ = 4.6 ns) with a spatial resolution of ~0.1 μm and 10 particles per cell. The temporal resolution, determined by the plasma frequency of the maximum density, was 0.046 fs (0.028 fs) for $t_{LFEX}$ = 2.6 ns ($t_{LFEX}$ = 4.6 ns). Reflection boundary condition was applied to the fields. Particles escaping from the simulation box were re-injected from the opposite boundary with a thermal

velocity. Fast electrons were observed at 1 μm beneath the rear surface of the plasma, which corresponded to ~140 μm from the target center for both simulation cases. To collect all the fast electrons, the simulation was run for 6 ps (5 ps) for $t_{LFEX}$ = 2.6 ns ($t_{LFEX}$ = 4.6 ns).

To eliminate the influence of the difference in laser intensities on the simulated results, another simulation was performed for the $t_{LFEX}$ = 4.6 ns case by using the same laser intensity as that for the $t_{LFEX}$ = 2.6 ns case (~$3.7 \times 10^{18}$ W cm$^{-2}$).

## Data availability
The data that support the findings of this study are available from the corresponding authors upon reasonable request.

## Code availability
Computer codes are available from the corresponding authors upon reasonable request.

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

## Acknowledgements
This work was supported by the Grants-in-Aid for Scientific Research (S) from the Japan Society for the Promotion of Science (JSPS) of Japan (Grant No. 15H05751) and the NIFS Collaboration Research Program (Grant Nos. NIFS15KUGK092, NIFS15KNSS060, and NIFS16KUGK103). The radiative hydrodynamic code used in this work was in part developed by the DOE NNSA-ASC OASCR Flash Center at the University of Chicago. The authors are thankful to H. Hosokawa for her great help in the target fabrication and to Dr. H. Sakagami for his great assistance in the PIC simulation. The authors also thank Dr. P. A. Norreys and Dr. L. Ceurvorst for helpful discussions.

## Author contributions
K.A.T. and H.H. designed and executed the experiment as principal investigators with help from T.G., K.S., Y.A., S.F., H.N., H.S. and M.S.W.; K.Abe, K.Aizawa, Y.E., Y.F., Y.H., A.I., S.K., S.L., H.M., T.Matsumoto, T.Minami, K.Okazaki, K.Okida, T.O., K.S., T.T. and M.Y. operated diagnostics; K.Y. and Y.I. fabricated the targets. Simulations were performed by T.G. for the FLASH modeling, M.Y. for the PIC modeling. Data analysis was carried out by T.G. and H.H. T.G. wrote the paper along with H.H. and K.A.T.; figures were prepared by T.G., Y.H., and M.Y.

## Competing interests
The authors declare no competing interests.
