## [Peer Review File · Nature Communications]

Reviewers' comments:

Reviewer #1 (Remarks to the Author):

The paper "Direct observation of imploded core heating via fast electrons with the super penetration scheme" reports on some interesting results that can be obtained in very few facilities in the world. However the experimental results and the analysis and simulations results present so much speculation and free parameters, that it is difficult to have a coherent idea of the physics going on. For this reason, even if I appreciate the paper's result and tentative analysis, I cannot recommend the paper for publication in Nature communication. More detailed comments follow, some of which could help improve the presentation of the paper if the authors wish to submit elsewhere their result.

1) The main objection is that there is no way to infer the core heating in the supposedly best scenario, that is the 2.6 ns delay. Namely In this case the Cu-K_{alpha} detection is the smallest and while the authors suggest three possible reasons (that the compressed core is smallest at this time, that the target opacity result in an intensity decay or that the core temperature enhancement reduces the emission), without a supplementary diagnostic or simulations allowing to verify if indeed one of this reason is the correct one, the results are not exploitable.

2) Along the same line, some interesting results are obtained with the electron spectrometer. However this result only allow to say that the population of detected electrons is too fast and cannot have participate to the heating so that from this measurement there is no direct indication of the population that should be absorbed by the core. As a result the authors make a hypothesis about the 'slow' electron population that should participate to the heating (number and temperature) that while in no contradiction with the data, is not really justified or proven either.

3) The main parameters of interest for the three shots should be given from the beginning. Only later on we discover they had different energies, the peak intensity of the UIL laser or that the relativistic correction to the critical density, although small, should be considered.

4) PIC simulations are performed to justify the behaviour of the heating in different cases, but it is not used to interpret the spectrometer result, while it should allow to get insight both in the fast and slow population.

Overall I think this paper has some interesting and encouraging results, but is not apt for Nature communication, the authors should submit to a more specialised journal, re-organising the material so that the supplementary information and methods are part of the text and the reader is not forced to go back and forth between the main figures and the supplementary figures, that are as important for the comprehension (for exemple the hydro profiles).

Reviewer #2 (Remarks to the Author):

This manuscript reports on the first experimental measurement of the energy deposited by short pulse induced fast electrons in a compressed sphere, approximating ICF conditions. This is an important step in revisiting the original fast ignitor scheme that does not use guiding cones which break the implosion symmetry. The experimental data is carefully analyzed to extract the efficiency of core heating by the short pulse. I recommend publication once some of the links between data and between data and simulations and resulting conclusions are clarified.

Fig. 4 caption should remind reader that curves are set by matching measured K-alpha yield.

For a broader audience, the authors need to tell readers the range of electron energies most important to heating the core, both for current case and 1.5 g/cm².

Since the authors argue through Fig. S5 that the divergence angle at 2.6 ns is 30-40°, and T1 assumed independent of angle per Fig. 4, the measured level of T1 electrons at 42° could be relevant and is it consistent with 0.8% coupling efficiency?

Given the various steps in estimating the energy coupling efficiency, the final result only deserves one significant figure ($0.8 \pm 0.3\%$).

Responses to Reviewers

We greatly appreciate the valuable comments from the reviewers that have made our manuscript more elaborate and clear. Our responses to these comments are shown below. As recognized in one of the reviewers' comments, we believe that our work should be published as a key milestone for fast ignition laser fusion in Nature Communications. The understanding of our study will benefit significantly for the progress not only for the laser fusion but also for the high energy density physics such as laboratory astrophysics and laser nuclear physics.

Responses to Reviewer #1

Comment #0:

The paper "Direct observation of imploded core heating via fast electrons with the super penetration scheme" reports on some interesting results that can be obtained in very few facilities in the world. However the experimental results and the analysis and simulations results present so much speculation and free parameters, that it is difficult to have a coherent idea of the physics going on. For this reason, even if I appreciate the paper's result and tentative analysis, I cannot recommend the paper for publication in Nature communication. More detailed comments follow, some of which could help improve the presentation of the paper if the authors wish to submit elsewhere their result.

- We can't agree that the speculation and free parameters in our analysis prevent the clarification of the physics. The speculation in our analysis is physically reasonable, while the free parameters actually have little effect on our primary conclusion. Therefore, we believe that our results and analysis have presented a clear physical picture as well as a reliable conclusion. More details follow below.

Comment #1:

The main objection is that there is no way to infer the core heating in the supposedly best scenario, that is the 2.6 ns delay. Namely in this case the Cu-K_{alpha} detection is the smallest and while the authors suggest three possible reasons (that the compressed core is smallest at this time, that the target opacity result in an intensity decay or that the core temperature enhancement reduces the emission), without a supplementary diagnostic

or simulations allowing to verify if indeed one of this reason is the correct one, the results are not exploitable.

- The reviewer might not be familiar with the research field of fast ignition. **In fact, core heating is clearly characterized by the Cu K-alpha emission and strongly justified by calculations that are based on the benchmarked radiative hydrodynamic simulations.**
- **Radiative hydrodynamic simulations were performed to justify our analysis and these simulations well reproduced the experimental results.** The simulations were benchmarked by the temporal target self-emission in photon energy from 1 keV to 10 keV. As shown in Supplementary Figure S1 on Page 27, the simulated time history of the peak emission agreed well with the experimental result. According to these radiative hydrodynamic simulations, the injection time of the short pulse at 2.6 ns corresponded to the peak compression of the target. Based on the simulated plasma parameters at 2.6 ns, the Cu K-alpha emission was calculated, which agreed well with the measurement. Both of them had a side-to-side diameter of ~100 micron meters, as shown in Figure 2 (b, e, h) on Page 8. This agreement demonstrated that our experiments and simulations are self-consistent.
- **The three reasons we suggest for the weak Cu K-alpha emission at 2.6 ns are physically reasonable and strongly justified by the good agreement between the calculation and the measurement.** **Firstly**, the fast electrons produced by the short pulse are instinctively angularly divergent, which is widely known in the community of fast ignition for inertial confinement fusion. That's why great efforts are being made to collimate the fast electrons so that more energy could be transported to the core region. Because of their angular divergence, fewer electrons are able to collide with the Cu contained region and hence to excite Cu K-alpha emission, as the target is compressed. Therefore, the small size of the core at 2.6 ns is one of the reasons for the weak Cu K-alpha emission. **Secondly**, according to our benchmarked radiative hydrodynamic simulations, the peak areal density at 2.6 ns is ~0.06 g/cm², which is much higher than those at 2.2 ns and 4.3 ns [see Figure 1 (b) on Page 6]. Moderate absorption of the Cu K-alpha line (~8 keV) through the Cu contained plastic is expected under this peak areal density due to the target opacity. **Thirdly**, the radiative hydrodynamic simulations indicate that the electron temperature at the core center at 2.6 ns is ~200 eV [see Supplementary Figure S3 (c) on Page 29]. As discussed by H. Sawada et al. [Physics of Plasmas 19, 103108 (2012)] and L. C. Jarrott et al. [Nature Physics 12, 499 (2016)], the enhancement of electron temperature could shift and broaden the Cu K-alpha line, and hence reducing the detection efficiency of the Cu K-alpha imager. **All these three factors have been taken into account in our**

calculations of the Cu K-alpha emission, which shows that the latter two factors respectively lead to intensity reduction by factors of 27% and 23% at the image center [see the last paragraph on Page 11]. In other words, if we neglect the target opacity and the reduction in the detection efficiency, the calculated Cu K-alpha intensity at the image center would be enhanced by a factor of $(1+27\%)*(1+23\%)=1.56$. However, we emphasize that our calculations are constrained by the absolute Cu K-alpha intensity from the HOPG spectrometer. It means that the relative intensity of the calculated Cu K-alpha images between different shots should be consistent with that of the measured K-alpha images. We find that these two are consistent only when the target opacity and the detection efficiency are considered [see Figure 2 on Page 8]; otherwise, the intensity of the calculated Cu K-alpha image at 2.6 ns would be overestimated. The good agreement between calculations and measurements shown in Figure 2 justifies our conclusion that the weak emission at 2.6 is due to the abovementioned three factors.

- **Fast electron induced core heating is well characterized by the Cu K-alpha emission.** Inferring core heating from the excited Cu K-alpha emission is a well-developed method and has been widely used in the research field of fast ignition. When the fast electrons are propagating through the Cu contained plasma, they excite Cu K-alpha emission and deposit energy by collision simultaneously. Because the emitted Cu K-alpha photons and the deposited energy display a similar dependence on the electron energy above 30 keV, the latter can be quantitatively inferred from the former. Similar work has been done by L. C. Jarrott et al. [Nature Physics 12, 499 (2016)] and S. Sakata et al. [Nature Communications 9, 3937 (2018)]. In our experiments, Cu K-alpha emission produced by fast electrons at the core region at 2.6 ns is clearly observed, indicating that fast electrons are indeed able to reach the core region and hence to heat the core by collision. With the aid of the absolute Cu K-alpha photons measured in experiments, the deposited energy at the time of peak compression is inferred, which corresponds to ~1% of the short pulse energy. Admittedly, the Cu K-alpha emission at the peak compression in our experiments is weak, which indicates that not too much energy is transported to the core by the fast electrons. However, it is undeniable that fast electrons have passed through the core and that core heating has taken place. We would like to emphasize that our focus in this paper is to report the first direct observation of core heating by fast electrons, rather than to find a way to enhance the energy coupling efficiency. The latter requires continuous efforts in the fast ignition community and will be a part of our future work.

Comment #2:

Along the same line, some interesting results are obtained with the

electron spectrometer. However this result only allow to say that the population of detected electrons is too fast and cannot have participate to the heating so that from this measurement there is no direct indication of the population that should be absorbed by the core. As a result the authors make a hypothesis about the ‘slow’ electron population that should participate to the heating (number and temperature) that while in no contradiction with the data, is not really justified or proven either.

- **The two-temperature feature (i.e. a ‘slow’ component and a ‘fast’ component) in the energy spectra of fast electrons has been widely observed in numerical simulations and similar experiments.** Examples can be found literatures written by A. Pukhov, et al. [Physics of Plasmas 5, 1880 (1998)], A. J. Kemp, et al. [Physical Review E 79, 066406 (2009)], L. C. Jarrott, et al. [Physics of Plasmas 21, 031211 (2014)], and S. Fujioka, et al. [Physical Review E 91, 063102 (2015); Physics of Plasmas 23, 056308 (2016)].
- **In our PIC simulations, the energy spectra of generated fast electrons also display a clear two-temperature feature, as shown below.**
- The ‘slow’ electron population was not measured in our experiments, primarily because of our integrated experimental configuration. When the low-energy electrons were passing through the compressed core and the surrounding corona, they were absorbed and deflected. We admit that the ‘slow’ electron population plays an important role in our experiments and needs careful measurement. However, its measurement should be made in a separate experiment rather than the integrated experiment reported here.

Figure 1. Energy spectra of fast electrons in PIC simulations. The red and blue curves represent the results for the $t_{LFEX} = 2.6 \text{ ns}$ and $t_{LFEX} = 4.6 \text{ ns}$ cases, respectively. A clear feature of two-temperature is observed in these simulated spectra.

Comment #3:

The main parameters of interest for the three shots should be given from the beginning. Only later on we discover they had different energies, the peak intensity of the UIL laser or that the relativistic correction to the critical density, although small, should be considered.

- We appreciate this suggestion. A table is added to list the main parameters in the joint shots on Page 6.

Comment #4:

PIC simulations are performed to justify the behaviour of the heating in different cases, but it is not used to interpret the spectrometer result, while it should allow to get insight both in the fast and slow population.

- **PIC simulations do allow to get some insight in the energy spectra of fast electrons. The simulated results are plotted in Figure S10 on Page 36 in the revised manuscript.**
- We note that our PIC simulation is unable to angularly resolve the energy spectra. The energy spectra shown above (Figure 1) are obtained by counting all the electrons escaping from the exiting boundary of the simulation box. Besides, due to the limitation of computational resources, the simulation is performed in 2D planar geometry, which is different from the 3D geometry in experiments. Therefore, one can not expect to compare the absolute quantities (e.g. population and slope temperature) between simulations and experiments. But the comparison of the relative tendency could be instructive. For instance, the simulated energy spectra display a clear two-temperature feature, which validates our assumption of the electron energy spectra in the analysis of the Cu K-alpha emission and the core heating. The simulations show a cooler temperature in the fast population at 4.6 ns than that at 2.6 ns, which is consistent with the experimental result shown in Supplementary Figure S4 on Page 30.

Responses to Reviewer #2

Comment #0:

This manuscript reports on the first experimental measurement of the energy deposited by short pulse induced fast electrons in a compressed sphere, approximating ICF conditions. This is an important step in revisiting the original fast ignitor scheme that does not use guiding cones which break the implosion symmetry. The experimental data is carefully

analyzed to extract the efficiency of core heating by the short pulse. I recommend publication once some of the links between data and between data and simulations and resulting conclusions are clarified.

- We thank the reviewer for his/her positive comments on our work. Our response to each of the specific comments follows below.

Comment #1:

Fig. 4 caption should remind reader that curves are set by matching measured K-alpha yield.

- A sentence is added in Fig. 4 caption (Page 13) as well as in Fig. 3 caption (Page 12).

Comment #2:

For a broader audience, the authors need to tell readers the range of electron energies most important to heating the core, both for current case and 1.5 g/cm².

- The optimum electron energy ranges for current case and 1.5 g/cm² are added on Page 13.

Comment #3:

Since the authors argue through Fig. S5 that the divergence angle at 2.6 ns is 30-40°, and T1 assumed independent of angle per Fig. 4, the measured level of T1 electrons at 42° could be relevant and is it consistent with 0.8% coupling efficiency?

- **The divergence angle of 30°~40° inferred from Fig. S5 is for fast electrons produced at 4.6 ns; while the coupling efficiency of 0.8% is estimated at the time of peak compression (2.6 ns). These two are not expected to be consistent with each other due to their different timings.** The divergence angle of fast electrons at 4.6 ns could be estimated because of the large size of the Cu K-alpha emission. At 2.6 ns, the time of peak compression, the Cu K-alpha emission is so small (~100 microns) that it could provide little information of angular divergence of fast electrons. But according to our PIC simulations, the divergence angle of fast electrons produced at 2.6 ns is larger than that at 4.6 ns due to the longer density scalelength at the former time, as discussed on Page 15. The coupling efficiency is estimated at 2.6 ns rather than at 4.6 ns, because the coupling at the time of peak

- compression (2.6 ns) is of most interest in fast ignition.
- One may further ask if the T1 electrons at 42° measured at 2.6 ns is consistent with the 0.8% coupling efficiency. Our answer is that the coupling efficiency of 0.8% is weakly dependent on the electrons at 42° and is generally valid in the parameter domain of interest. At 2.6 ns, the core is ~100 microns in diameter and the fast electrons are produced ~160 microns away from the core center. As a result, fast electrons with a propagating angle larger than 17° can not collide with the core. Of course, the data at 42° provide some information of the T1 electrons. But these data alone can not allow to infer either the angular distribution or the energy spectra of fast electrons passing through the core. Note that, in reality, the slope temperature measured at 42° could be different from that at 0° (on-axis). That's why we introduce two free parameters (divergence angle and slope temperature T1) to describe the T1 electrons. As shown by our calculations, the coupling efficiency is strongly related to the absolute Cu K-alpha photons and is weakly dependent on either the angular divergence or the slope temperature. Therefore, the coupling efficiency is generally valid in the parameter domain of interest.

Comment #4:

Given the various steps in estimating the energy coupling efficiency, the final result only deserves one significant figure ($0.8 \pm 0.3\%$).

- The final energy coupling efficiency ($0.83 \pm 0.34\%$) is replaced by ($0.8 \pm 0.3\%$). (Page 5 and Page 12)

Reviewers' comments:

Reviewer #1 (Remarks to the Author):

Revised version of the manuscript

"Direct observation of imploded core heating via fast electrons with super-penetration scheme".

The resubmitted manuscript is slightly improved, however I still feel that the way the manuscript is presented is not optimal. In particular the author's answer to most of the questions is not very satisfactory. In no way I contest K_α heating as a diagnostic of core heating, or the fact that the 2.6 shot was for the highest compression. Nor I am contesting the fact that you typically have a two temperature population.

What I am contesting in the paper is that different models and parameters are stretched to fit the data, without being fully coherent among themselves. However as I already said there is no real contradiction either. It would maybe be useful for the reader at some point to have a summary of all the hypothesis used to match the data for the shot at 2.6 with with respect to the spectrometer and the K_α : which two temperatures (within interval or error bars) have been used, what is the electron flux or density, which part is supposed to be absorbed and which pass, divergence angle, etc.

Once the author have done this and shown explicitly that while there has been a choice parameters, but this choice is fully coherent, I consider that the manuscript will be apt for publication.

Reviewer #2 (Remarks to the Author):

The authors have satisfactorily responded to comments and added requested clarifications.

Responses to Reviewers

The authors appreciate the reviewers' time and effort for reviewing our manuscript. The initial version of the manuscript was written in the style of another Nature journal. We have reformatted it thoroughly by adding section and subsection titles, so that it adapts to Nature Communications and displays a clearer text structure. The manuscript has also been revised by taking into account the reviewers' comments. We hope that the revised manuscript has presented our work in a better way.

Responses to Reviewer #1

Comments:

The resubmitted manuscript is slightly improved, however I still feel that the way the manuscript is presented is not optimal. In particular the author's answer to most of the questions is not very satisfactory. In no way I contest K_{α} heating as a diagnostic of core heating, or the fact that the 2.6 shot was for the highest compression. Nor I am contesting the fact that you typically have a two temperature population.

What I am contesting in the paper is that different models and parameters are stretched to fit the data, without being fully coherent among themselves. However as I already said there is no real contradiction either. It would maybe be useful for the reader at some point to have a summary of all the hypothesis used to match the data for the shot at 2.6 with with respect to the spectrometer and the K_{α} : which two temperatures (within interval or error bars) have been used, what is the electron flux or density, which part is supposed to be absorbed and which pass, divergence angle, etc.

Once the author have done this and shown explicitly that while there has been a choice parameters, but this choice is fully coherent, I consider that the manuscript will be apt for publication.

Responses from the authors:

The authors are grateful to the reviewer for his/her constructive comments, which have helped us improve the manuscript a lot. The manuscript is thoroughly revised in accordance with these comments. More details follow below.

We have moved some hydrodynamic simulation results from the Supplementary Information to the main text as suggested by the reviewer in his/her first round of review, so as to improve the coherence of the manuscript. Specifically, the target self-emission and the profiles of both density and temperature are added in Subsection II B, as shown in Figure 2 (page 8) and Figure 3 (page 10), respectively. The accompanying text follows Figure 2, as highlighted on page 9. The rest figures still remain in the Supplementary Information so that the main text is focused on our primary conclusion.

To give the readers a summary of all the hypotheses used in our data analysis, two paragraphs are added at the beginning of Subsection II C, as highlighted in pages 12-14. The first paragraph introduces the physics in the derivation of the deposited energy from the Cu K-alpha emission. While the second one introduces the reasons why fast electrons consist of two component with different slope temperatures and why the deposited energy can be derived even without knowing the exact values of T_1 and θ_1 .

A table is also added in Subsection II C (Table 2 on page 13) to present the parameters of the high-temperature (T_2) component of fast electrons, including the slope temperature (T_2), the divergence angle (θ_2), the population (N_2), and the total energy (E_2). These parameters are obtained by fitting the experimentally measured electron spectra. They are also the parameters used in the eTrans simulations to fit the Cu K-alpha emission. The data listed in this table demonstrate that the T_2 -component fast electrons carry minimal energy and contribute little both to the Cu K-alpha emission and to the core heating.

The slope temperatures (T_1) and the divergence angles (θ_1) of the T_1 -component fast electrons used in the eTrans simulations are summarized in Figure 5 (page 15). We use the total energy rather than the flux or the density to characterize the fast electron beam, because the latter two are space-dependent quantities for an angularly distributed fast electron beam. The total energy of fast electrons is further divided by the short pulse energy, so as to obtain the laser-to-electron energy conversion efficiency, which is of more interest in the research field of fast ignition. A subfigure is added in Figure 5 to present the laser-to-electron energy conversion efficiencies obtained in the eTrans simulations. The text accompanying this subfigure is highlighted on pages 15-16.

The kinetic energy of fast electrons optimal for heating the core

(0.06 g/cm²) in our experiment ranges from 200 keV to 500 keV. We have stated it in Section III, as highlighted on page 17.

We hope that our revisions listed above have made our manuscript fully coherent.

REVIEWERS' COMMENTS:

Reviewer #1 (Remarks to the Author):

The authors took into account previous remarks and clarified a number of points in their presentation. In the current form the manuscript it's easier to read and the results more accessible. I consider that it is now apt to publication in Nature Communications